# VQKV: High-Fidelity and High-Ratio Cache Compression via Vector-Quantization

## Abstract

The increasing context length in Large Language Models (LLMs) leads to a proportional growth of the Key-Value (KV) cache, posing a significant challenge for their deployment in resource-limited settings. While existing training-free methods for KV cache compression, such as token eviction, feature dimension reduction, and scalar quantization, can reduce memory usage, they often do so at the cost of diminished model performance, especially at high compression ratios. To resolve the trade-off between memory efficiency and model fidelity, we introduce VQKV, a novel, training-free KV cache compression method based on vector quantization. Instead of discarding tokens or compressing individual dimensions, VQKV maps entire high-dimensional cache vectors to a compact, learned codebook. This approach allows for the representation of thousands of floating-point values with just a few integer indices corresponding to the codebook. As a result, VQKV achieves a significant compression ratio while enabling high-fidelity reconstruction of the original cache vectors through a simple codebook lookup. VQKV achieves a high compression ratio with minimal performance degradation. Extensive evaluations on LLaMA3.1-8B and LLaMA3.2-3B models across long-context benchmarks demonstrate that VQKV significantly outperforms existing state-of-the-art compression methods at similar compression ratios, highlighting its effectiveness in preserving information while substantially reducing the memory footprint of the KV cache.

## 1 Introduction

Large Language Model (LLM) has already found widespread applications in many fields due to their outstanding capabilities. However, the scaling of context length results in continuous growth of the Key-Value (KV) cache, which in turn limits the feasibility of employing large language models in resource-constrained environments. Although approaches such as sparse attention and parameter quantization can effectively alleviate memory usage, they frequently incur a degradation in model performance. Therefore, developing a memory-efficient method that can simultaneously preserve model performance is of critical importance for addressing this bottleneck in large language models.

Among existing training-free methods for KV cache compression, three main categories can be identified: token eviction, feature dimension compression, and scalar quantization. Token eviction methods (Li et al., 2024b; Zhang et al., 2023b; Xiao et al., 2023) reduce the KV cache size by selectively discarding less critical token representations. While this strategy effectively shortens the sequence length to achieve high compression ratios, it incurs an irreversible loss of information, which can impair performance on tasks requiring long-range contextual understanding. Feature dimension compression techniques (Chang et al., 2024; Yuan et al., 2023; Liu et al., 2024a) exploit the inherent redundancy within the high-dimensional KV vectors, often through methods like low-rank decomposition. These approaches store a compressed representation and reconstruct the full vector as needed. However, they often face a trade-off between compression ratio and performance, as the low-rank approximation can struggle to retain all necessary information, particularly at higher compression levels. Scalar quantization (Liu et al., 2024c; Yuan et al., 2023; Hooper et al., 2024) is another widely used technique that reduces memory usage by independently compressing each floating-point value in the cache to a lower bit-width. This approach faces a significant challenge in maintaining fidelity at high compression ratios. By treating each feature independently, it fails

to exploit the correlations and structural information within the high-dimensional vectors. Consequently, pushing to very low bit-widths (e.g., 2 or 3 bits) often introduces substantial quantization error, leading to a sharp decline in model performance.

In summary, these existing methods consistently struggle to balance between memory efficiency and model performance. A natural question arises: Can we achieve high compression by preserving the most salient information rather than discarding tokens or compressing individual dimensions? To answer this, we propose **VQKV**, a novel training-free KV cache compression method using vector quantization (VQ). VQKV maps high-dimensional cache vectors to a compact codebook, storing only the corresponding integer indices to replace the original cache. This joint quantization captures the intrinsic structure of the data, a key advantage over scalar quantization that treats each dimension in isolation. During attention, the original cache is reconstructed via a simple codebook lookup. By preserving vector-level information, VQKV achieves substantial memory savings with a high compression ratio, while maintaining high fidelity and model performance across various downstream tasks.

Extensive evaluations on LLaMA3.1-8B (Dubey et al., 2024) and LLaMA3.2-3B (AI, 2024) across the LongBench (Bai et al., 2023), NIAH (Li et al., 2024a), and RULER (Hsieh et al., 2024) benchmarks demonstrate that our VQKV significantly outperforms existing methods at comparable ratios. Remarkably, on some tasks, VQKV even surpasses the performance of the uncompressed full-cache baseline. Our contributions can be summarized as follows:

- We introduce vector quantization (VQ) to the problem of KV cache compression. This novel perspective fundamentally addresses the limitations of existing methods by preserving the intrinsic structure and correlations within the KV vectors.

- We design a simple yet high effective training-free compression framework, VQKV. This method reduces the memory footprint of the KV cache during decoding at the cost of a modest amount of additional computation, thereby enabling the model to handle longer contexts on resource-constrained devices.

- We conduct extensive experiments demonstrating SOTA performance. Through evaluations of LLaMA3.1-8B (Dubey et al., 2024) and LLaMA3.2-3B (AI, 2024) on long-context benchmarks (LongBench (Bai et al., 2023), NIAH (Li et al., 2024a) and RULER (Hsieh et al., 2024)), we show that VQKV consistently outperforms existing training-free compression methods at comparable ratios.

## 2  RELATED WORK

**Cache Compression**  Previous work such as MLA (Liu et al., 2024a) and CSKV (Wang et al., 2024), though achieving a high compression ratio by projecting high-dimensional caches into low-dimension vectors, often need continue pretraining on LLMs or even need training LLMs from scratch, which consumes a lot of computation. Beyond these training-based approaches, methods like StreamingLLM (Xiao et al., 2023), H2O (Zhang et al., 2023b), SnapKV (Li et al., 2024b) and PyramidKV (Cai et al., 2024) alleviate the bottlenecks in memory by selectively retaining or structurally organizing the KV representations of important tokens, thereby reducing the memory usage while preserving generation quality. But evicting token is not always robust because of unavoidable information loss, especially at the view of long context retrieval tasks. Applying scalar quantization directly on feature dimension of KV cache can substantially reduce memory usage while improving computational efficiency (Liu et al., 2024c; Chang et al., 2024; Hooper et al., 2024). However, the low precision in the retained cache leads to a decline in model performance. To sum up, existing training-free efforts on KV cache compression often lack the exploration on feature dimension, or suffer from insufficient precision when reconstructing the cache.

**Vector Quantization**  Vector quantization is a widely recognized technique known for its effective compression and representation abilities. Wav2Vec (Baevski et al., 2020) and HuBERT (Hsu et al., 2021) use VQ to extract discrete pseudo labels from continuous raw waves for unsupervised learning. SoundStream (Zeghidour et al., 2021) introduces residual skill to VQ, thus enriching the representation space of codebooks. In computer vision, VQ is generally used for concentrating information from pixels (He et al., 2022; Van Den Oord et al., 2017; Esser et al., 2021; Zheng et al.,

Figure 1: Overview of our VQKV. The left part shows the detailed process of our VQKV on prefilling stage and decoding stage. The right part shows the overview of our VQKV.

2022). At the same time, VQ helps introduce information from other modalities into LLMs (Liu et al., 2024b; Zhang et al., 2023a). Other works use VQ for constraining the specialized formats of generation, such as code generation (Liu et al., 2025b) and action generation (Wang et al., 2025). Previous work have shown great compression and reconstruction capability of VQ for continuous vectors. However, VQ has not been applied in representing text features on LLM nor in compressing caches. Our VQKV first introduce VQ to represent context information and gain great efficacy.

## 3 METHODOLOGY

VQKV employs VQ to compress the KV cache along the feature dimension. By leveraging separate codebooks for the key and value caches, VQKV achieves high-fidelity reconstruction while substantially reducing memory consumption, which is particularly beneficial for handling long-context scenarios. In this section, we first describe how VQ is learned for compressing the KV cache, followed by a detailed discussion of how VQKV is integrated into the inference process.

### 3.1 LEARNING VQ FOR KV CACHE

We use Residual Simple Vector Quantization (RSimVQ) for KV cache compression. Combining residual skill (Zeghidour et al., 2021) with SimVQ (Zhu et al., 2024), RSimVQ consists of multiple codebooks and has an extra projection matrix on every codebook to enhance expressiveness and utilization. Each codebook can represent an independent subspace of original cache by an index of the nearest codebook entry. Then, RSimVQ sums up all the codebook entries retrieved from codebooks to reconstruct the original cache.

Specifically speaking, let $x$ be the original cache and $\hat{x}$ be the reconstructed cache, $Q_i$ be the codebook with entries $q_1, ..., q_{N_c}$ where $N_c$ is the codebook size, and $z$ be the selected index. For each codebook, RSimVQ first finds the nearest entry and records corresponding index $z$.

$$\hat{x} = \operatorname{argmin}_{q \in \{q_1, ..., q_{N_c}\}} \|x - Wq\| \triangleq Wq_z \tag{1}$$

where $W$ is the projection matrix of each codebook. Then, RSimVQ sends the residual part $(x - \hat{x})$ to $Q_{i+1}$ for next step quantization.

$$x \leftarrow x - \hat{x}, \quad z_i \leftarrow z \tag{2}$$

Iteratively, RSimVQ could map the original cache into a few set of codes $z_1, ..., z_{N_q}$. VQKV only store codebooks and these quantized codes in the memory instead of the original cache vector, leading the memory usage of original KV cache from $[L, D]$ floats (ignoring layer and head dimension here for simplicity) into $[L, N_q]$ integers, which would reduce a lot of memory footprint when the cache vectors are massive. When reconstructing, RSimVQ simply looks up entries according to these codes and sums these entries up.

$$\hat{x} = \sum_i^{N_q} Q_i[z_i] \tag{3}$$

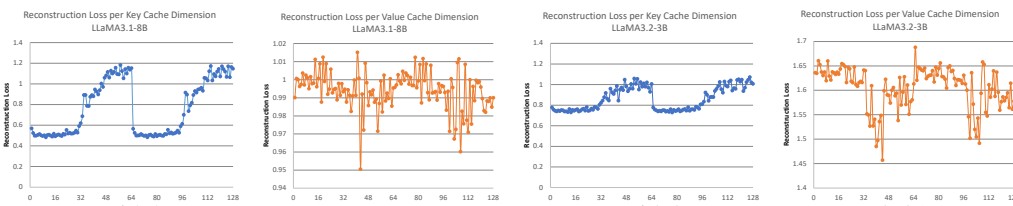

Figure 2: Reconstruction loss per dimension of KV cache on OpenWebText (Gokaslan et al., 2019). The positions of key cache dimensions with poor reconstruction exhibit a periodic pattern, leading us to separate low- and high-frequency components and use two independent RSimVQ on reconstructing key cache.

We prefetch the KV cache from around 10M tokens to train all these codebooks and two independent RSimVQs are trained separately for key and value cache. More details about training are described in Section 4.1. During training, RSimVQ uses a stop-gradient operation $\text{sg}(\cdot)$ to keep gradient flowing after discretization, along with the training loss as follows

$$\mathcal{L} = \|x - \hat{x}\|^2 + \beta\|q_z - \text{sg}(x)\|^2 + \gamma\|x - \text{sg}(q_z)\| \tag{4}$$

After training, we evaluate the cache reconstruction capability of RSimVQ on OpenWebText (Gokaslan et al., 2019) and observe that the positions of key cache dimensions with poor reconstruction exhibit a periodic pattern. Figure 2 shows that when building the vector spaces of key cache, VQ tends to overfitting on some head dimensions while underfitting on other dimensions. This observation indicates that different dimensions of key caches vary in different reconstruction difficulty. Similarly, Liu et al. (2025a) has also reported the same anomalous phenomenon. We attribute this to the Rotary Positional Embedding (RoPE)(Su et al., 2024) mechanism in the model: the lower dimensions of the key cache encode low-frequency information while the higher dimensions encode high-frequency information, leading to the heterogeneous distribution of the key cache. These additional position information prevent a single RSimVQ from adequately reconstructing the entire key cache. To address this issue, we partition the key cache dimensions into low- and high-frequency components according to the reconstruction quality, and employ two independent RSimVQs to reconstruct them separately.

## 3.2 USING VQKV ON INFERENCE TIME

Figure 1 illustrates the overall workflow of VQKV, where the **left** side depicts the compression–reconstruction process of the KV cache, and the **right** side presents how VQKV is integrated into the prefill and decoding stages. Specifically, during **prefill**, VQKV compresses each KV cache vector by mapping it to the nearest entries in multiple codebooks and storing only the corresponding indices as KV codes. During **decoding**, VQKV compresses the cache of each new token, updates the stored KV codes, and reconstructs the required KV cache from the codebooks, while maintaining a local sliding window by discarding the oldest entries.

To be more specifically, in the prefilling stage, we preserve both an initial segment of the KV cache of length $L_{init}$ and the most recent segment of length $L_{local}$ from compression.

$$K^i, K^l = K[: L_{init}], K[-L_{local} :]; \quad V^i, V^l = V[: L_{init}], V[-L_{local} :] \tag{5}$$

Then we distinguish the low- and high-frequency components in key cache according to the reconstruction quality. $D^l, D^h$ are the corresponding dimensions of the two components. And we have the intermediate KV cache to be compressed:

$$K^{ml} = K[L_{init} : -L_{local}, D^l], \quad K^{mh} = K[L_{init} : -L_{local}, D^h]$$
$$V^m = V[L_{init} : -L_{local}] \tag{6}$$

As discussed above, we use three trained RSimVQs to compress intermediate cache $K^{ml}, K^{mh}, V^m$ into KV codes $K^{cl}, K^{ch}, V^c$ correspondingly. Let the codebook number of each RSimVQ be

$N_q^l, N_q^h, N_q^v$, then the compressed KV cache have the size $K^{cl} \in \mathbb{N}^{L \times N_q^l}, K^{ch} \in \mathbb{N}^{L \times N_q^h}, V^c \in \mathbb{N}^{L \times N_q^v}$. After compressing, we use the **original** KV for forward propagation.

$$K^{cl} = \mathcal{VQ}^l(K^{ml}), \quad K^{ch} = \mathcal{VQ}^h(K^{mh}), \quad V^c = \mathcal{VQ}^v(V^m) \tag{7}$$

$$O = \text{attention}(Q, K, V) \tag{8}$$

In the decoding stage, we compress the cache out of the local range individually.

$$K^{cl} \leftarrow K^{cl} + \mathcal{VQ}^l(K^l[0, D^l]), \quad K^{ch} \leftarrow K^{ch} + \mathcal{VQ}^h(K^l[0, D^h])$$
$$V^c \leftarrow V^c + \mathcal{VQ}^v(V^l[0]) \tag{9}$$

Then VQKV reconstructs the intermediate KV cache by selecting the entries from each codebook according to the stored indices.

$$\hat{K}^m[D^l] = \mathcal{VQ}^{-1}(K^{cl}), \quad \hat{K}^m[D^h] = \mathcal{VQ}^{-1}(K^{ch}), \quad \hat{K} = \text{cat}(K^i, \hat{K}^m, K^l)$$
$$\hat{V}^m = \mathcal{VQ}[V^c], \quad \hat{V} = \text{cat}(V^i, \hat{V}^m, V^l) \tag{10}$$

With the current query $q_{t+1}$, we calculate the attention output $o_{t+1}$ by

$$o_{t+1} = \text{attention}(q_{t+1}, \hat{K}, \hat{V}) \tag{11}$$

The total compression ratio of VQKV is associated with the number of codebooks. For KV cache with dimension of $D^K$ and $D^V$, our VQKV achieves a compression ratio $r$ such that

$$r = \left(1 - \frac{N_q^v + N_q^l + N_q^h}{D^K + D^V}\right) \times 100\% \tag{12}$$

### 3.3 Efficiency Optimzation

To improve the efficiency of the quantization process, we optimize the algorithm for computing the nearest distances within codebooks. By adopting a block-wise computation strategy, we effectively reduce the peak memory consumption during distance calculations. Moreover, since the residual structure in RSimVQ inherently lacks parallelism, we enhance parallel efficiency in the quantization process by performing batched quantization computations based on the current $L_{local}$. Specifically, instead of quantizing KV cache one by one during LLM decoding, we compute the quantization process on the entire cache segment $K^l$ and $V^l$ in a single operation.

$$K^{cl} \leftarrow K^{cl} + \mathcal{VQ}^l(K^l[:, D^l]), \quad K^{ch} \leftarrow K^{ch} + \mathcal{VQ}^h(K^l[:, D^h])$$
$$V^c \leftarrow V^c + \mathcal{VQ}^v(V^l) \tag{13}$$

In this way, the compression step of VQ is performed only once every $L_{local}$ steps, which would otherwise occur at every decoding step, thereby substantially improving efficiency. When reconstructing the cache, we only rebuild the portion required for the current decoding step.

In addition, since VQKV reconstructed the intact attention portion, our VQKV is natively compatible with acceleration framework like FlashAttention (Dao et al., 2022; Dao, 2024) and vLLM (Kwon et al., 2023). Moreover, the decompression process of VQKV can be integrated into standard FlashDecoding, allowing the compressed KV codes to be progressively decoded during the computation of every blocks of the sequence, thereby further reducing memory consumption and improving time efficiency. It is worth noting that this functionality has not been implemented in the experiments reported in this paper; nevertheless, our approach still achieves a substantial reduction in memory usage. In future work, we plan to optimize VQKV on FlashDecoding (Dao, 2024) with customized triton kernel to gain better memory and latency efficiency.

## 4 Experiment

### 4.1 Setup

We conduct all our experiments on LLaMA3.1-8B (Dubey et al., 2024) and LLaMA3.2-3B (AI, 2024). For both models, we set the length of initial token $L_{init}$ to 4 and the length of local tokens $L_{local}$ to 1024. We sample 0.1% data of OpenWebText (Gokaslan et al., 2019)

for RSimVQ training and every RSimVQ is trained with learning rate 0.001 and batch size 65536. The size of codebooks vary from cache type and models. For LLaMA3.1-8B (Dubey et al., 2024), we use the codebook number $(N_q^v, N_q^l, N_q^h) = (8, 20, 16)$ with the codebook size $(N_c^v, N_c^l, N_c^h) = (65536, 65536, 16384)$, while for LLaMA3.2-3B (AI, 2024), we use a codebook number $(N_q^v, N_q^l, N_q^h) = (10, 20, 14)$ and set all codebook sizes to 65536. By Equation 12, our method achieves a compression ratio of 82.8% on both model. As discussed in Section 3.3, all our experiments are launched without FlashAttention (Dao et al., 2022; Dao, 2024).

## 4.2 Long-Context Evaluation

The long-context evaluation experiments are all performed on multiple NVIDIA H100 GPU with FP32 precision. We evaluate our VQKV against other KV cache optimization methods with three long-context benchmarks on OpemCompass (Contributors, 2023): LongBench (Bai et al., 2023), Needle-In-A-Haystack(NIAH) (Li et al., 2024a) and RULER (Hsieh et al., 2024). All tasks are set with a truncation context length of 32K. We compare with the ASVD (Yuan et al., 2023), SnapKV (Li et al., 2024b), Palu (Chang et al., 2024) and KIVI (Liu et al., 2024c). For fair comparison, we set the ratio parameters to 0.2 in ASVD, keep 2048 recalled middle tokens in SnapKV, retain 70% of the KV in Palu and apply 4-bit quantization, 75% compression in KIVI. The results of SnapKV and Palu on NIAH are referred from Liu et al. (2025a)

Across LongBench (Table 1), NIAH (Figure 3, 4), and RULER (Figure 5 and Table 2), our VQKV consistently achieves the best trade-off between compression and performance. On LongBench, it delivers average scores closest to the uncompressed baseline and surpasses existing baselines under both LLaMA3.1-8B and LLaMA3.2-3B. On NIAH, our approach maintains a perfect 100 score, identical to the full-cache model, while other methods exhibit clear degradation. On RULER, it preserves strong long-context capability, achieving results close to the baseline and substantially outperforming competing methods, even at 32K context length. These results collectively demonstrate the robustness and effectiveness of our approach in both general and long-context scenarios.

| | Single-Doc | | | Multi-Doc | | | Summary | | | Few-shot | | | Synthetic | | Code | | Avg. |
|---|---|---|---|---|---|---|---|---|---|---|---|---|---|---|---|---|---|
| | NQ | Qsp | MF | HQ | WQ | Msq | GR | QS | MN | TR | TQ | SS | PC | PR | LCC | Re-P | |
| *LLaMA3.1-8B* | 12.9 | 20.2 | 32.4 | 12.0 | 14.0 | 8.7 | 29.8 | 25.2 | 1.0 | 73.5 | 91.0 | 47.2 | 0.8 | 26.8 | 72.2 | 69.2 | 33.6 |
| + ASVD | 4.6 | 9.9 | 16.1 | 9.7 | 7.4 | 5.2 | 9.0 | 16.6 | 12.8 | 60.0 | 78.7 | 33.7 | 2.8 | 4.9 | 30.4 | 36.9 | 21.2 |
| + SnapKV | 12.7 | 19.8 | 32.5 | 12.0 | 13.8 | 8.6 | 29.2 | 24.9 | 12.6 | 73.0 | 91.0 | 46.5 | 0.8 | 26.8 | 60.0 | 59.7 | 32.7 |
| + Palu | 6.4 | 16.6 | 23.1 | 9.7 | 12.3 | 6.6 | 16.5 | 21.5 | 10.7 | 72.5 | 84.6 | 37.1 | 1.3 | 14.3 | 64.7 | 59.1 | 28.6 |
| + KIVI | 19.2 | 20.8 | 31.2 | 15.2 | 17.7 | 8.2 | 17.6 | 12.1 | 3.9 | 72.5 | 89.6 | 33.9 | 2.2 | 15.9 | 67.6 | 64.4 | 30.7 |
| + **Ours** | 13.4 | 19.7 | 30.6 | 11.4 | 13.8 | 8.2 | 26.1 | 23.9 | 0.9 | 73.0 | 91.3 | 46.1 | 0.8 | 31.8 | 71.5 | 68.9 | **33.2** |
| *LLaMA3.2-3B* | 10.3 | 21.6 | 34.9 | 9.7 | 13.0 | 6.8 | 30.2 | 23.7 | 28.2 | 70.0 | 87.2 | 38.2 | 0.0 | 7.0 | 70.0 | 66.4 | 32.3 |
| + ASVD | 0.8 | 10.7 | 8.8 | 5.4 | 5.2 | 2.7 | 8.7 | 8.6 | 11.1 | 31.5 | 49.1 | 17.7 | 3.3 | 3.5 | 35.4 | 36.6 | 14.9 |
| + SnapKV | 6.0 | 21.5 | 33.5 | 9.9 | 12.9 | 7.1 | 22.8 | 24.1 | 27.1 | 65.0 | 87.2 | 38.4 | 0.0 | 7.0 | 69.6 | 64.3 | 31.0 |
| + Palu | 12.2 | 16.7 | 27.7 | 9.9 | 11.6 | 6.6 | 24.2 | 20.0 | 19.4 | 58.0 | 86.4 | 41.4 | 0.3 | 4.7 | 56.2 | 54.3 | 28.1 |
| + KIVI | 7.4 | 22.1 | 33.1 | 9.8 | 12.0 | 5.2 | 17.4 | 14.2 | 16.0 | 69.5 | 88.3 | 30.3 | 1.2 | 7.4 | 70.0 | 65.6 | 29.3 |
| + **Ours** | 11.7 | 21.3 | 35.3 | 10.0 | 14.4 | 7.0 | 29.2 | 23.2 | 27.3 | 70.0 | 87.2 | 38.8 | 0.0 | 7.0 | 70.3 | 64.8 | **32.3** |

Table 1: Results of LLaMA3.1-8B (Dubey et al., 2024) and LLaMA3.2-3B (AI, 2024) on Long-Bench (Bai et al., 2023). Our VQKV achieves the closest performance to the full cache models on comparable compression ratio against other methods.

## 4.3 Memory Efficiency

To show the memory efficiency of our VQKV, we test the maximum generation length of LLaMA3.1-8B on a single NVIDIA A100 40GB with FP32 without FlashAttention. As shown in Table 3, our method maintains nearly the same peak memory usage as the baseline across different generation lengths, with only a marginal increase. More importantly, it significantly extends the maximum generation length: while the baseline LLaMA3.1-8B encounters an out-of-memory error at around 25k tokens, our method supports over 52k tokens on a single NVIDIA A100 40GB GPU, doubling the usable context length. These results demonstrate that our approach effectively pre-

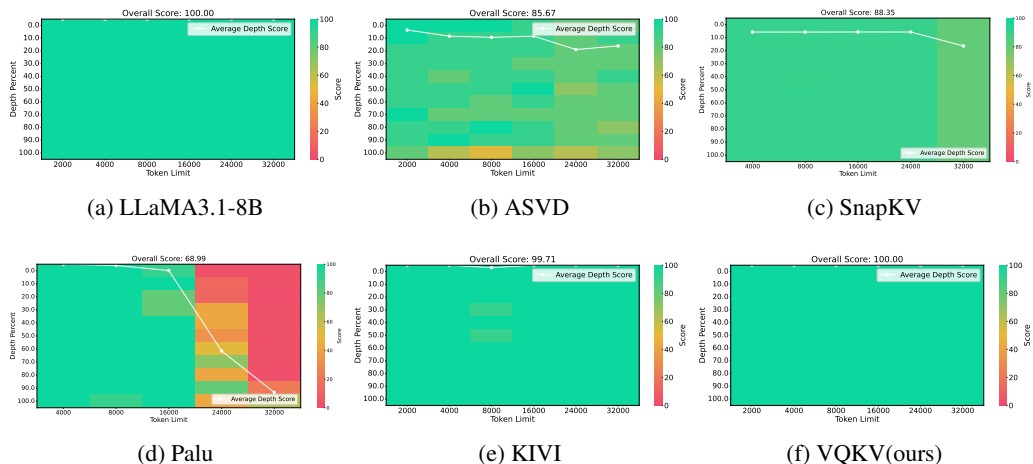

Figure 3: Results of LLaMA3.1-8B (Dubey et al., 2024) on Needle-In-A-Haystack (NIAH)(Li et al., 2024a). The results of SnapKV (Li et al., 2024b) and Palu (Chang et al., 2024) are referred from Liu et al. (2025a). Our VQKV maintain a perfect score while other methods exhibit clear degradation.

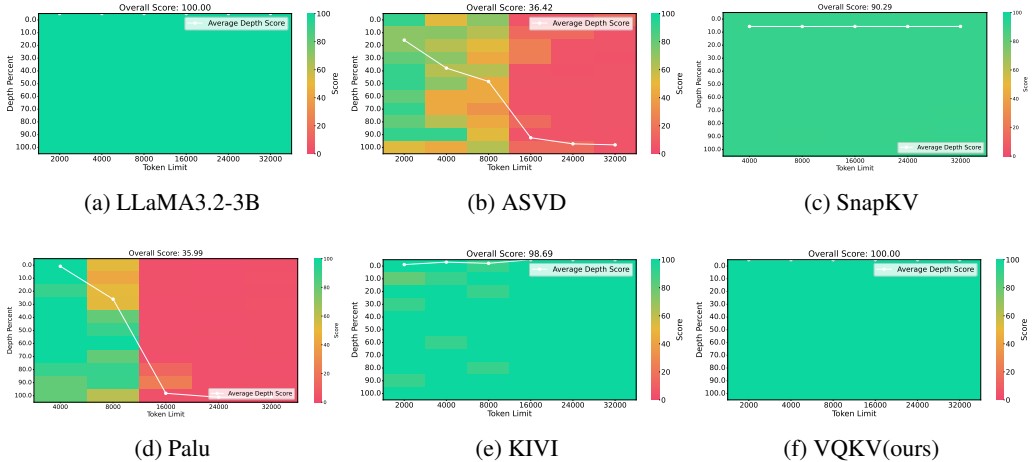

Figure 4: Results of LLaMA3.2-3B (AI, 2024) on Needle-In-A-Haystack (NIAH)(Li et al., 2024a). The results of SnapKV (Li et al., 2024b) and Palu (Chang et al., 2024) are referred from Liu et al. (2025a). Our VQKV maintains a perfect score while other methods exhibit clear degradation.

|  | 4K | 8K | 16K | 32K | Avg. |
|---|---|---|---|---|---|
| *LLaMA3.1-8B* | 94.74 | 92.78 | 93.12 | 89.59 | 92.56 |
| + ASVD | 39.75 | 32.94 | 25.27 | 19.29 | 29.31 |
| + SnapKV | 91.18 | 77.12 | 69.80 | 58.52 | 74.16 |
| + Palu | 74.70 | 66.01 | 59.80 | 52.44 | 63.24 |
| + KIVI | 57.91 | 52.81 | 47.60 | 49.73 | 52.01 |
| **+ Ours** | 93.96 | 89.80 | 87.47 | 79.10 | **87.58** |
| *LLaMA3.2-3B* | 90.14 | 85.69 | 82.91 | 78.15 | 84.22 |
| + ASVD | 27.15 | 20.83 | 15.95 | 9.51 | 18.36 |
| + SnapKV | 85.33 | 69.57 | 61.73 | 58.74 | 68.84 |
| + Palu | 71.75 | 65.38 | 59.69 | 55.08 | 62.98 |
| + KIVI | 37.92 | 49.37 | 41.41 | 36.28 | 41.25 |
| **+ Ours** | 88.50 | 79.31 | 73.75 | 67.14 | **77.18** |

Table 2: Results of LLaMA3.1-8B (Dubey et al., 2024) and LLaMA3.2-3B (AI, 2024) on RULER (Hsieh et al., 2024). Our VQKV outperforms other methods on 4K, 8K, 16K and 64K length.

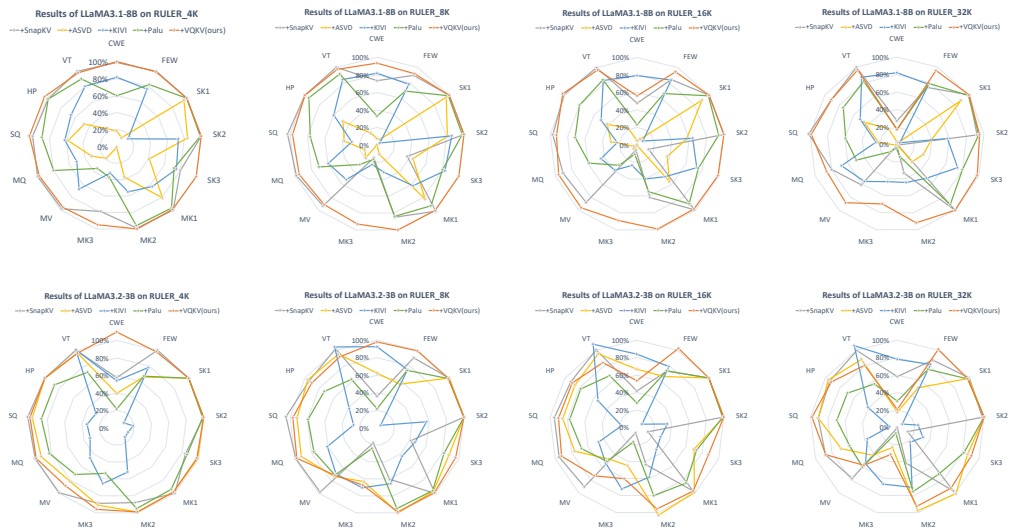

Figure 5: Detailed results of LLaMA3.1-8B (Dubey et al., 2024) and LLaMA3.2-3B (AI, 2024) on RULER (Hsieh et al., 2024) in different context length. The results of full cache model are taken as the 100% reference.

serves memory efficiency while substantially enhancing long-sequence generation capability, highlighting its advantages in both compression ratio and memory footprint.

| | Generation Length | | | | | | | | | | Max Length |
|---|---|---|---|---|---|---|---|---|---|---|---|
| | 32 | 128 | 256 | 512 | 1024 | 2048 | 4096 | 8192 | 16384 | 32768 | |
| *LLaMA3.1-8B* | 29.94 | 29.96 | 30.02 | 30.07 | 30.21 | 30.49 | 31.05 | 32.18 | 34.43 | OOM | 25841 |
| + Ours | 30.63 | 30.66 | 30.71 | 30.76 | 31.74 | 31.83 | 32.02 | 32.38 | 33.44 | 35.66 | 52096 |

Table 3: Peak memory usage in generation time on LLaMA3.1-8B (Dubey et al., 2024). Our method achieves twice longer generation length than full-cached base model on a single NVIDIA A100 40GB, demonstrating great compression ratio and memory footprint of VQKV.

# 5 ABLATION STUDY

We evaluate our approach on different combination of codebook numbers and codebook sizes. On Longbench (Bai et al., 2023), NIAH (Li et al., 2024a) and RULER (Hsieh et al., 2024), the ablation on codebook numbers show that more codebooks contribute to better performance but lower compression ratio. Bigger codebook sizes also help with performance improvement but hurt the memory footprint.

As shown in Table 4, when the total number of parameters of the codebooks is comparable, increasing the number of codebooks tends to yield a more substantial improvement in downstream performance. However, an excessively large number of codebooks directly reduces the compression ratio, in which case larger codebook sizes are required to ensure high-fidelity reconstruction.

The observed phenomenon can be explained as follows. Increasing the number of codebooks effectively enhances the representational capacity of the quantization process. Each additional codebook provides an extra subspace in which the input vector can be approximated, thereby distributing the representational burden. In residual or compositional quantization schemes, this corresponds to a multi-stage refinement where successive codebooks progressively correct the residual error, leading to a substantial reduction in quantization distortion.

| Codebook Number | | | Codebook Size | | | Ratio | Param | LongBench | NIAH | RULER |
|---|---|---|---|---|---|---|---|---|---|---|
| V | L | H | V | L | H | | | | | |
| 12 | 30 | 24 | | | | 74.2% | 277.27 | 33.30 | 100.00 | 90.25 |
| 10 | 25 | 20 | | | | 78.5% | 231.05 | 33.45 | 100.00 | 89.21 |
| 8 | 20 | 16 | 64k | 64k | 16k | 82.8% | 184.85 | 33.21 | 100.00 | 87.58 |
| 6 | 15 | 12 | | | | 87.1% | 138.63 | 32.40 | 99.86 | 77.38 |
| 4 | 10 | 8 | | | | 91.4% | 92.42 | 30.82 | 83.60 | 39.01 |
| | | | 64k | 64k | 64k | 82.8% | 222.60 | 33.23 | 100.00 | 87.77 |
| | | | 64k | 64k | 32k | 82.8% | 184.85 | 33.21 | 100.00 | 87.58 |
| 8 | 20 | 16 | 32k | 32k | 32k | 82.8% | 55.87 | 32.57 | 100.00 | 84.71 |
| | | | 4k | 4k | 4k | 82.8% | 14.19 | 32.03 | 99.71 | 76.93 |
| | | | 1k | 1k | 1k | 82.8% | 3.77 | 31.61 | 95.05 | 61.08 |

Table 4: Ablation on codebook numbers and codebook sizes. We evaluate different combinations on LongBench (Bai et al., 2023), NIAH (Li et al., 2024a), and RULER (Hsieh et al., 2024). Detailed results are in Appendix B.

In contrast, enlarging the size of a single codebook primarily improves the granularity of representation within that codebook. While larger codebooks allow for finer partitioning of the input space, the benefit diminishes due to the inherent challenges of high-dimensional vector spaces: newly added entries may not efficiently capture the distribution of inputs, resulting in limited marginal gains.

Therefore, increasing the number of codebooks is akin to adding additional layers of expressive power, which yields a more pronounced improvement in reconstruction fidelity, whereas enlarging the codebook size merely increases the resolution of each layer and provides relatively modest benefits.

# 6 CONCLUSION

In this paper, we proposed **VQKV**, a training-free KV cache compression framework that leverages vector quantization to jointly capture correlations within cache vectors. By replacing continuous representations with compact discrete codes, VQKV achieves high compression ratios while preserving fidelity, thus overcoming the limitations of token eviction, feature dimension compression, and scalar quantization. Extensive experiments on LLaMA3.1-8B and LLaMA3.2-3B show that VQKV consistently outperforms existing training-free methods and even surpasses the full-cache baseline in some cases, demonstrating its effectiveness for memory-efficient long-context inference.

# 7 LIMITATIONS

Our method still has room for improvement in decoding efficiency. In principle, it can be integrated with FlashDecoding and further accelerated with Triton kernels to enhance efficiency and reduce memory consumption. However, such optimizations are not pursued in this paper, and we leave them for future work.

# 8 ETHICAL STATEMENT

This study adheres to recognized ethical guidelines and professional standards. It does not involve human participants, handle sensitive personal information, or engage in applications with potential ethical concerns. All experimental procedures and analyses were conducted in accordance with established norms, ensuring scientific rigor, transparency, and reliability.

## 9 REPRODUCIBILITY STATEMENT

To ensure the reproducibility of this paper, we will publicly release our VQKV, the complete training and inference code and all checkpoints used in our experiments. We expect these as a reference for efficient LLM improvement, motivating and advancing more progress in this field.

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

## A  THE USE OF LARGE LANGUAGE MODELS

This paper uses LLMs only for polishing writing.

## B  DETAILED RESULT ON ABLATION STUDY

We conduct ablation studies on the number and size of codebooks. On LLaMA3.1-8B, we evaluate different combinations of codebook numbers and codebook sizes, and test them on LongBench, NIAH, and RULER. The detailed experimental results are presented as follows.

| ID | Codebook Number | | | Codebook Size | | | Compression Ratio | Total Param |
|----|------|------|------|------|------|------|------|------|
|    | V | L | H | V | L | H | | |
| **A** | 12 | 30 | 24 |  |  |  | 74.2% | 277.27 |
| **B** | 10 | 25 | 20 |  |  |  | 78.5% | 231.05 |
| **C** | 8 | 20 | 16 | 64k | 64k | 16k | 82.8% | 184.85 |
| **D** | 6 | 15 | 12 |  |  |  | 87.1% | 138.63 |
| **E** | 4 | 10 | 8 |  |  |  | 91.4% | 92.42 |
| **F** |  |  |  | 64k | 64k | 64k | 82.8% | 222.6 |
| **G** |  |  |  | 64k | 64k | 16k | 82.8% | 184.85 |
| **H** | 8 | 20 | 16 | 16k | 16k | 16k | 82.8% | 55.87 |
| **I** |  |  |  | 4k | 4k | 4k | 82.8% | 14.19 |
| **J** |  |  |  | 1k | 1k | 1k | 82.8% | 3.77 |

Table 5: Experiment settings for ablation study. For simplification, every combination of codebooks is marked with a letter as experiment ID.

| ID | Single-Doc | | | Multi-Doc | | | Summary | | | Few-shot | | | Synthetic | | Code | | Avg. |
|---|---|---|---|---|---|---|---|---|---|---|---|---|---|---|---|---|---|
| | NQ | Qsp | MF | HQ | WQ | Msq | GR | QS | MN | TR | TQ | SS | PC | PR | LCC | Re-P | |
| A | 12.7 | 20.2 | 31.8 | 11.2 | 14.5 | 8.2 | 28.7 | 24.5 | 0.9 | 73.5 | 91.0 | 46.1 | 0.8 | 27.3 | 72.2 | 69.3 | 33.3 |
| B | 13.7 | 20.6 | 30.1 | 11.7 | 14.3 | 8.2 | 28.4 | 25.6 | 0.9 | 73.5 | 91.0 | 46.1 | 0.8 | 29.3 | 72.1 | 69.1 | 33.5 |
| C | 13.4 | 19.7 | 30.6 | 11.4 | 13.8 | 8.2 | 26.1 | 23.9 | 0.9 | 73.0 | 91.3 | 46.1 | 0.8 | 31.8 | 71.5 | 68.9 | 33.2 |
| D | 12.8 | 17.6 | 28.2 | 11.7 | 13.8 | 8.3 | 20.4 | 24.2 | 0.6 | 71.5 | 91.0 | 45.6 | 0.8 | 32.8 | 71.3 | 67.9 | 32.4 |
| E | 13.0 | 15.7 | 26.2 | 11.4 | 14.2 | 7.9 | 16.7 | 22.9 | 0.6 | 59.0 | 91.0 | 44.7 | 0.6 | 31.3 | 71.0 | 67.1 | 30.8 |
| F | 13.3 | 20.0 | 30.1 | 11.6 | 13.8 | 8.2 | 26.3 | 24.6 | 0.8 | 73.5 | 91.3 | 45.7 | 0.8 | 31.3 | 71.6 | 68.9 | 33.2 |
| C | 13.4 | 19.7 | 30.6 | 11.4 | 13.8 | 8.2 | 26.1 | 23.9 | 0.9 | 73.0 | 91.3 | 46.1 | 0.8 | 31.8 | 71.5 | 68.9 | 33.2 |
| G | 11.9 | 18.9 | 29.0 | 11.2 | 13.3 | 8.2 | 23.3 | 23.1 | 0.9 | 73.5 | 90.8 | 46.1 | 0.8 | 29.8 | 71.9 | 68.5 | 32.6 |
| H | 12.9 | 18.9 | 27.8 | 11.7 | 13.5 | 8.0 | 20.5 | 23.3 | 0.8 | 71.0 | 90.8 | 45.3 | 0.8 | 28.0 | 71.3 | 68.1 | 32.0 |
| I | 14.4 | 17.0 | 27.6 | 11.6 | 14.1 | 7.9 | 17.9 | 23.1 | 0.7 | 67.0 | 90.9 | 44.4 | 0.8 | 29.8 | 71.2 | 67.5 | 31.6 |

Table 6: Results of different numbers and sizes of codebook on LongBench.

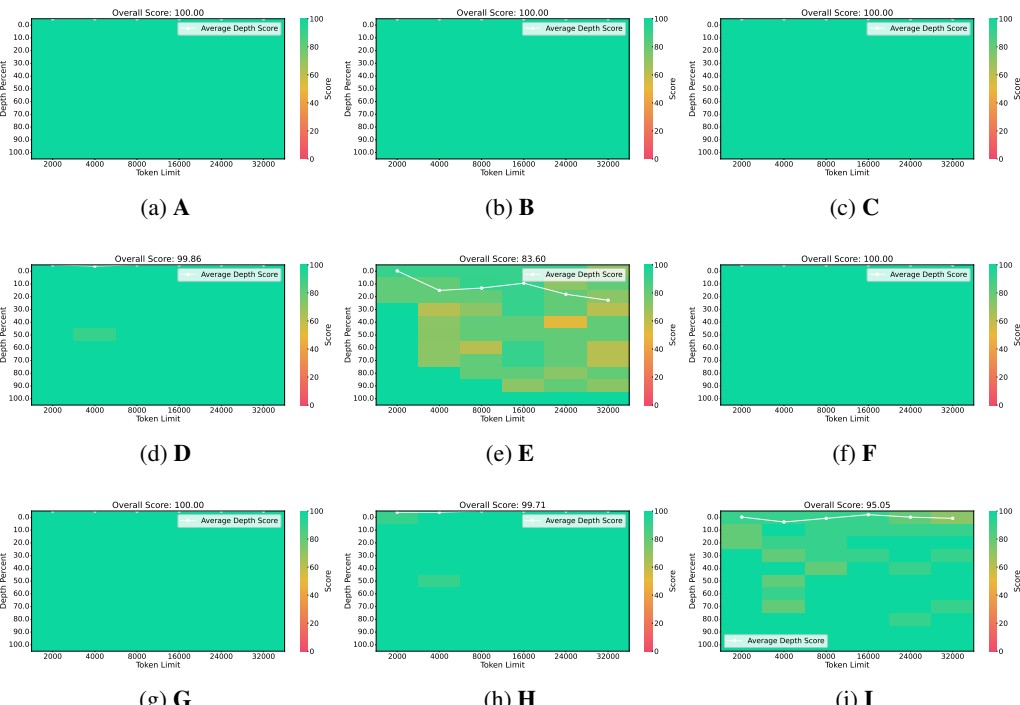

Figure 6: Results of different numbers and sizes of codebook on Needle-In-A-Haystack.

| ID | 4K | 8K | 16K | 32K | Avg. |
|---|---|---|---|---|---|
| A | 94.42 | 91.69 | 90.47 | 84.40 | 90.25 |
| B | 94.04 | 90.89 | 88.77 | 83.13 | 89.21 |
| C | 93.96 | 89.80 | 87.47 | 79.10 | 87.58 |
| D | 87.86 | 79.38 | 75.06 | 67.22 | 77.38 |
| E | 56.51 | 41.09 | 32.46 | 25.96 | 39.01 |
| F | 93.76 | 90.21 | 87.20 | 79.90 | 87.77 |
| C | 93.96 | 89.80 | 87.47 | 79.10 | 87.58 |
| G | 91.84 | 87.11 | 83.54 | 76.36 | 84.71 |
| H | 86.56 | 79.99 | 74.53 | 66.63 | 76.93 |
| I | 74.59 | 64.77 | 57.16 | 47.81 | 61.08 |

Table 7: Results of different numbers and sizes of codebook on RULER

