# OpenReview forum: "VQKV: High-Fidelity and High-Ratio Cache Compression via Vector-Quantization"
_ICLR.cc/2026/Conference — ICLR 2026 Conference Withdrawn Submission_

### Official Review · Reviewer_HaSW · 2025-10-28

**Soundness:** 2
**Presentation:** 3
**Contribution:** 2
**Rating:** 2
**Confidence:** 4

**Summary:**

This paper introduces VQKV, a novel method to compress the KV cache in LLMs using vector quantization. Instead of discarding tokens or compressing individual values, VQKV maps entire high-dimensional cache vectors to a compact, learned codebook, storing only integer indices. Evaluations on LLaMA3 models show VQKV outperforming selected baselines on long-context benchmarks.

**Strengths:**

1. The evaluation results are good. VQKV evaluated on 3 well established long-context benchmark and got good performance at a high compression ratio.

**Weaknesses:**

1. The novelty is somewhat limited. Vector quantization on LLMs has already been explored in previous literatures. For example, on weights(QuiP#[1], AQLM[2]) or KV cache(VQLLM[3], CommVQ[4]).
2. Lack of efficiency evaluation. A major goal for compressing KV cache is to reduce the fetching time from GPU HBM(which is the bottleneck of decoding latency), while VQKV still has to fetch the entire KV cache from HBM thus likely provides no efficiency gain.  Moreover, the paper does not discuss the efficiency part and provides no evaluation on the efficiency gain (loss) or analyze the reconstruction overhead.

[1] QuIP#: Even Better LLM Quantization with Hadamard Incoherence and Lattice Codebooks

[2] Extreme Compression of Large Language Models via Additive Quantization

[3] Residual vector quantization for KV cache compression in large language model

[4] CommVQ: Commutative Vector Quantization for KV Cache Compression

**Questions:**

Could the authors provide a quantitative analysis of the latency overhead? Specifically, what is the impact on throughput (e.g., tokens/second) for VQKV compared to the full-cache baseline and a fast scalar quantization baseline like KIVI, especially at long context lengths?

---

### Official Review · Reviewer_DQdM · 2025-10-29

**Soundness:** 2
**Presentation:** 2
**Contribution:** 2
**Rating:** 2
**Confidence:** 5

**Summary:**

The paper proposes a KV-cache compression method based on vector quantization. The authors observe that Key caches are more sensitive to quantization error than Value caches and mitigate this by using two codebooks for Keys, aiming to preserve accuracy while achieving high compression.

**Strengths:**

* Applies vector quantization to increase the compression ratio of the KV cache, targeting substantial memory footprint reduction.
* Keys are more sensitive to quantization error than Values, so the method introduces two codebooks for the Key cache to mitigate this issue.

**Weaknesses:**

- The main concern is end-to-end throughput in real systems. While the paper reports memory savings (compression ratios), it provides no latency/throughput evaluation. In memory-bound LLM inference, reduced footprint can correlate with higher throughput, but this is not guaranteed. Practical speedups depend on kernel design, cache behavior, and data movement.
- Table 4 lists codebook configurations, but efficient dequantization typically requires the codebooks to reside in a cache that can be read with low latency. The reported codebook sizes seem large. It is unclear whether they fit the target on-chip caches, raising questions about lookup overheads.
- Experiments are limited to Llama-3. Demonstrating generality on other families (e.g., Gemma, Qwen, Phi, etc.) would strengthen the claims.
- Minor: In Table 1, bolding the best score per task would improve readability.
- Minor: Table 3 uses FP32. Most LLM inference uses FP16/BF16. Please justify the FP32 choice or provide FP16/BF16 results.
- Minor: The compression ratios per method in Table 1 are not clearly specified. Is the target a uniform ~75% reduction vs. FP16? Please make this explicit.

**Questions:**

- Do the proposed codebooks fit in the intended GPU/accelerator cache (please specify cache or shared memory size)? Can you provide a latency estimate or microbenchmark to support expected end-to-end gains?
- Can you report results on models beyond Llama-3 (e.g., Gemma, Qwen, Phi) to demonstrate robustness of the approach?

---

### Official Review · Reviewer_qwrb · 2025-10-30

**Soundness:** 2
**Presentation:** 2
**Contribution:** 1
**Rating:** 2
**Confidence:** 5

**Summary:**

This paper proposes VQKV, a training-free method for KV cache compression in LLMs based on vector quantization. The method uses a residual vector-quantization scheme with separate codebooks for key and value caches. Considering the existence of RoPE, the authors further partition the key cache dimensions into low- and high-frequency components according to the reconstruction quality, and employ two independent codebooks to reconstruct them separately. They also focus on optimizing the efficiency of their algorithm and propose several techniques, such as a block-wise computation strategy and performing batched quantization computations. Experiments on long context benchmarks show the effectiveness of their method compared to normal KV cache quantization methods.

**Strengths:**

1. Their experiments are well presented and compared to lots of baselines, showing the effectiveness of vector quantization for KV cache.

2. Apart from quantization quality, the authors also discuss the efficiency of their method, which is an appropriate and important consideration. Since the primary motivation for quantizing the KV cache is to reduce memory usage and latency, the method is only meaningful if the computational overhead introduced by vector quantization does not outweigh the latency savings gained from reduced memory access.

**Weaknesses:**

I believe this work still has a lot of room for improvement and is currently not good enough, based on these reasons:

1. The claimed main contribution, vector quantization, is already broadly explored by prior works [1,2,3] and is not novel. The authors have made some improvements, for example, to apply two codebooks for low- and high-frequency components of the key cache, considering the impact of RoPE, though this improvement is not critical and is not well justified. For example, is it better to apply a single but twice larger codebook to the key cache?

2. The experimental results are not convincing. The authors do not compare their approach against other vector quantization methods, which raises questions about the actual effectiveness of the proposed technique. Moreover, the paper lacks a clear description of the quantization bit-width used for each method in the comparison, which is a crucial detail, since a fair evaluation requires comparable quantization settings.

3. Finally, the efficiency of the proposed method is not well justified. As I said in the above section, the method is only meaningful if the computational overhead introduced by vector quantization does not outweigh the latency savings gained from reduced memory access. I am concerned that vector quantization will induce a lot of computation as it requires a lot of summation in order to reconstruct the kv cache back, and this reconstruction needs to happen for every decoding step. The tricks that the authors claim can improve the efficiency are not critical and do not change the fact that the dequantization process is computationally extensive. Also, the claim that "our VQKV is natively compatible with acceleration frameworks like FlashAttention and vllm" is not well supported and seems to overclaim. There are no latency/throughput experiments presented in the paper to prove the real latency benefit of their proposed method.



[1] KV Cache is 1 Bit Per Channel: Efficient Large Language Model Inference with Coupled Quantization

[2] Residual vector quantization for KV cache compression in large language model

[3] CommVQ: Commutative Vector Quantization for KV Cache Compression

**Questions:**

1. Computation Overhead is the main issue for vector quantization. How does your method approach this? Can you present some results on latency?

2. The codebook size used is very large, up to 64k, considering that most samples in longbench have tokens less than 64k. Why do we need such a large codebook? And how does the codebook size compare to the KV cache size?

3. What new insights and contributions does the proposed method offer to the community compared with existing KV cache vector quantization methods?

---

### Official Review · Reviewer_4u4G · 2025-11-02

**Soundness:** 2
**Presentation:** 3
**Contribution:** 3
**Rating:** 4
**Confidence:** 3

**Summary:**

The paper proposes VQKV, a KV-cache compression framework that applies vector quantization (VQ) to keys and values. It learns multiple residual codebooks (RSimVQ) offline and stores only integer indices per token during decoding; reconstruction uses codebook lookups. The authors further split key dimensions into “low-/high-frequency” groups (attributed to RoPE) and quantize them with two independent VQs. On LLaMA-3.1-8B and LLaMA-3.2-3B, VQKV reports strong accuracy at high nominal compression, sometimes near full-cache quality, and extends maximum context on a single A100-40GB.

**Strengths:**

1. Applying vector quantization at the vector level (vs. per-scalar or token eviction) is a reasonable design to preserve intra-vector structure.

2. Insightful observation on periodic reconstruction errors in key dimensions, leading to a two-branch VQ for low/high “frequency” subspaces.

3. Ablations on codebook number/size clarify the capacity–compression trade-off and provide tuning guidance.

**Weaknesses:**

1. The method trains RSimVQ codebooks on ~10M tokens; this is not training-free which can affect the paper positioning.

2. Reconstruction adds matmul/lookups; decoding quantization is batched every $L_{\text{local}}$ but still incurs periodic overhead. There is no wall-clock comparison vs. widely-used baselines under FlashAttention-2/vLLM pipelines.

3. Baseline methods are fixed to particular ratios/knobs (e.g., 4-bit KIVI, SnapKV middle-token recall) that may not match VQKV’s effective memory/computation envelope. It’s unclear whether baselines were tuned for their Pareto front under the same context/truncation/runtime constraints.

**Questions:**

1. Please clarify the trianing-free claim.

2. Please report wall-clock latency and tokens/s vs. KIVI/SnapKV/ASVD under the same hardware and with FlashAttention-2/vLLM.

3. It will be better to provide full memory accounting: codebook parameters (dtype, per-layer/per-head replication), index storage (bits/index), and runtime working buffers.

---

### Note · Authors · 2025-12-04

I have read and agree with the venue's withdrawal policy on behalf of myself and my co-authors.